# Sterol Migration during Rotational Frying of Food Products in Modified Rapeseed and Soybean Oils

**DOI:** 10.3390/biom14030269

**Published:** 2024-02-23

**Authors:** Magdalena Rudzińska, Anna Gramza-Michałowska, Monika Radzimirska-Graczyk, Eliza Gruczyńska-Sękowska

**Affiliations:** 1Faculty of Food Science and Nutrition, Poznań University of Life Sciences, 60-637 Poznań, Poland; anna.gramza@up.poznan.pl; 2Faculty of Health Sciences, Poznan University of Physical Education, 61-871 Poznań, Poland; graczyk@awf.poznan.pl; 3Department of Chemistry, Institute of Food Sciences, Warsaw University of Life Sciences, 02-776 Warszawa, Poland; eliza_gruczynska@sggw.edu.pl

**Keywords:** plant oils, frying, cholesterol, phytosterols, degradation, migration, gas chromatography

## Abstract

This study explores the impact of rotational frying of three different food products on degradation of sterols, as well as their migration between frying oils and food. The research addresses a gap in the existing literature, which primarily focuses on changes in fat during the frying of single food items, providing limited information on the interaction of sterols from the frying medium with those from the food product. The frying was conducted at 185 ± 5 °C for up to 10 days where French fries, battered chicken, and fish sticks were fried in succession. The sterol content was determined by Gas Chromatography. This research is the first to highlight the influence of the type of oil on sterol degradation in both oils and food. Notably, sterols were found to be most stable when food products were fried in high-oleic low-linolenic rapeseed oil (HOLLRO). High-oleic soybean oil (HOSO) exhibited higher sterol degradation than high-oleic rapeseed oil (HORO). It was proven that cholesterol from fried chicken and fish sticks did not transfer to the fried oils or French fries. Despite initially having the highest sterol content in fish, the lowest sterol amount was recorded in fried fish, suggesting rapid degradation, possibly due to prefrying in oil with a high sterol content, regardless of the medium used.

## 1. Introduction

Frying is one of the most popular methods of preparing food for consumption, both at home and in industry. Deep-fat fried food products have grown in popularity, despite health trends that aim to decrease the amount of fat in meals. Fried food products have a pleasant taste and smell and a crunchy texture, which leads to high consumer acceptance. The market sizing and forecasts indicate that the fried chicken market is poised to grow by USD 1.92 billion during 2022–2026, progressing at a Compound Annual Growth Rate (CAGR) of 5.32%. Total world fisheries and aquaculture production showed a 45 percent growth between 2000 and 2021, reaching 182 million tonnes in 2021 and setting a new production record. The market for French fries was valued at USD 16.68 billion in 2022 and is expected to grow at a CAGR of 5.1% over the forecast period [1]. The deep-fried food market is segmented into fresh and frozen products, with prefried French fries, battered chicken, and fish being the most important frozen products. Many factors affect the success of the deep-fat frying process. One of the most important is the use of an appropriate frying medium. Consumers’ effect on the quality of the final product can thus be limited when they purchase prefried frozen products. The fat medium can also be used cyclically (repeatedly) to fry different products, such as chicken, fish, and potatoes.

Sterols are components of food lipids that play an important role in the human body. Cholesterol, a sterol of animal origin, is present in almost all cells of the human body and plays both good and bad roles. Cholesterol forms the backbone of all steroid hormones and vitamin D analogs. It is responsible for growth and development throughout life and may be useful as an anticancer facilitator. Because humans have a limited ability to catabolize cholesterol, it readily accumulates in the body when there is an excess of it from the diet or due to a genetic abnormality. This accumulation results in the foremost cause of death and disease, atherosclerosis. In contrast, a deficiency of cholesterol in the circulation may result in an inability to distribute vitamins K and E to vital organs, with serious consequences [2]. Cholesterol can also be oxidized to form a range of derivatives such as oxysterols, dimers, oligomers, volatile compounds, and other degraded compounds. Recent developments have shown that oxysterols can be involved in neurodegenerative diseases, especially Huntington’s disease, Parkinson’s disease, and Alzheimer’s disease, as well as in cancer and cardiovascular diseases [3].

On the other hand, there are plant sterols (called phytosterols), which reduce the uptake of cholesterol in the intestinal lumen and affect its transport, but which also regulate the metabolism of cholesterol in the liver. In addition, phytosterols can significantly reduce the plasma concentration of total cholesterol, triglycerides, and low-density lipoprotein cholesterol (LDL-C), with a dose–response relationship. Phytosterols can also activate the liver X receptor α-CPY7A1-mediated bile acid excretion pathway and accelerate the transformation and metabolism of cholesterol [4]. Phytosterols play a wide range of other roles, such as providing antioxidant and antiproliferative functions [5], anti-inflammatory and antipyretic effects [6], and hormone-like effects [7]. Both cholesterol and phytosterols are present in fats used for frying and in fried foods. It has been repeatedly demonstrated that cholesterol and plant sterols can be degraded by heat at different temperatures.

The degradation and oxidation of cholesterol when held at 150 °C has been studied by Chien et al. [8]. When meats and meat products were pan fried in rapeseed oil for ten minutes, cholesterol oxidation products were observed [9]. The protective role of conjugated linoleic acids (CLA) on cholesterol oxidation when held at 100 °C and 150 °C in a model system has shown the formation of oxysterols [10]. The effects of boiling and frying on the oxidation of cholesterol in processed foods have also been investigated [11]. Additionally, changes in phytosterol contents in fried oils and food products have been studied. For example, the transfer of bioactive compounds such as phytosterols during the frying of potatoes in olive oil shows that sterols are remarkably stable and that over 70% of them are retained in the frying of oil [12]. In a batch frying experiment, phytosterol loss ranged from 1% to 15%, and was highest in corn oil, followed by soybean oil and hydrogenated soybean oil [13].

The effects of cyclic frying of products with different sterol compositions have not yet been studied. In previous studies, French fries, chicken, and fish were rotationally fried in different plant oils for six or eleven days and the fatty acid composition, polar compounds, tocopherols, color, and sensory assessment were determined [14,15], though sterol changes and migration were not considered. The main goal of the present work was to study the sterol migration and degradation during rotational frying of French fries, battered chicken, and fish sticks in high-oleic rapeseed oil (HORO), high-oleic low-linolenic rapeseed oil (HOLLRO), soybean oil (SO), and high-oleic soybean oil (HOSO). Prefried French fries, battered chicken, and fish stocks were fried in succession, forming one rotational cycle, and nine cycles were run daily in each oil.

## 2. Materials and Methods

### 2.1. Food Products and Oils

Frozen French fries that had been prefried in high-oleic low-linolenic rapeseed oil and battered chicken and fish sticks prefried in refined rapeseed oil were used in the frying experiments. The same batch of frozen French fries, battered chicken, and fish sticks used in the experiments were purchased in local markets (Lethbridge, AB, Canada).

Four oils were used for frying: high-oleic rapeseed oil (HORO), high-oleic low-linolenic rapeseed oil (HOLLRO), refined soybean oil (SO), and high-oleic soybean oil (HOSO). All oils were commercially refined, bleached, and deodorized and obtained from Richardson Oil Processing (Lethbridge, AB, Canada).

### 2.2. Frying Procedure

The frying was performed at 185 ± 5 °C for ten days in stainless steel fryers (General Electric Company, New York, NY, USA). A total of 3.5 L of oil was placed in fryer and every second day 0.5 L of fresh oil was added. The oils were heated to 185 °C and held there for 2 h before frying. French fries, chicken, and fish were fried in each cycle, and nine cycles were run daily in each oil. A total of 400 g of French fries was fried for 5 min, and the same amount of chicken and fish was fried for 7 min. Each day, 3.6 kg of each product was fried, which gave 3.1 kg of food products per 1 L of oil.

### 2.3. Standards and Reagents

Standards of sterols, 5α-cholestane, and anhydrous pyridine were purchased from Sigma-Aldrich (Munich, Germany) and Merck (Darmstadt, Germany). The silylation mixture of BSTFA (N,O-bis(trimethylsilyl) trifluoroacetamide) with 1% TMCS (trimethylchlorosilane) was purchased from Fluka Chemie (Buchs, Switzerland). Other solvents and reagents were supplied by Sigma-Aldrich (Munich, Germany) and were of analytical or HPLC grade.

### 2.4. Extraction of Oils

The extraction of oils from food products was performed according to a method of Folch et al. [16] using methanol/chloroform (1:2, *v*/*v*). After separation, the solvent fraction was evaporated under nitrogen. The resulting lipids were stored at −20 °C for no longer than ten days in a freezer, Whirlpool—WSZ57L18DM (Casinetta, Italy).

### 2.5. Sterol Determination

The sterol content was determined according to the AOCS Official Method Ch 6-91 [17]. Extracted oils were saponified with 1M methanolic KOH for 18 h. Sterols were then extracted with hexane/methyl *tert*-butyl ether (1:1, *v*/*v*). The extracts were evaporated in a nitrogen stream and pyridine and BSTFA + 1% TMCS were added to the dry residues. For separation, an HP 6890 series II Plus (Hewlett Packard, Palo Alto, CA, USA) was used, equipped with a DB-35MS capillary column (25 m 0.20 mm, 0.33 m; J&W Scientific, Folsom, CA, USA). A sample of 1.0 µL was injected in splitless mode. The oven temperature was programmed to 100 °C for 5 min, then increased to 250 °C at 25 °C/min, held for 1 min, and then further increased to 290 °C at 3 °C/min, where it was held for 20 min. The carrier gas was hydrogen (1.5 mL/min). Sterols were identified by comparison of retention data with standards. 5α-Cholestane was used as internal standard. All analyses were performed in triplicate.

### 2.6. Statistical Analysis

The results are presented as means from three replicates ± standard deviations (SD). The statistical software Statistica 13.3 (Statsoft, Tulsa, OK, USA) was used to prepare a one-way analysis of variance (ANOVA). Probability values of less than 5% (*p* < 0.05) were considered to be significant.

## 3. Results

### 3.1. Oils

Sterol content and composition are presented in Figure 1. The highest sterol content was found in HORO, with 5.1 mg/g oil. In HOLLRO, the sterol level was also higher than in soybean oils, at 4.7 mg/g oil. The sterol level in soybean oils was much lower, at 1.9 mg/g for SO and 1.6 mg/g for HOSO. All fresh oils used for frying contained typical plant sterols. Brassicasterol, which is characteristic of the oil of Brassicaceae plants, was identified in both rapeseed oils. In contrast, stigmasterol content was found to be higher in soybean oils than in rapeseed oils. 

Sterol levels in HORO decreased to 4.6 mg/g after five days of frying and remained at that level after ten days of frying. For HOLLRO, there was no degradation of sterols during frying, with the total content remaining 4.3 mg/g. The sterol content of both rapeseed oils should be considered stable in this cyclic frying process. The content of sterols in both soybean oils was similar, and significantly increased to 3.4–3.7 mg/g after five and ten days of frying. This is due to the migration of sterols from the fat used to prefry the products before freezing.

Brassicasterol appeared in fried soybean oils at a level of 0.2–0.3 mg/g, indicating that the fried products were prefried in rapeseed oil. The level of avenasterol also increased fourfold in SO after frying, and was 0.1 mg/g in HOSO, which it was not detected prior to frying. In HOLLRO, the level of avenasterol increased by more than 40% after frying, and only in HORO did its level decrease after five and ten days of frying, by 7% and 9%, respectively.

These results indicate that the level of sterols in oils during cyclic frying is significantly affected by the type of fat used for prefrying.

### 3.2. Frozen Prefried French Fries

The content and composition of sterols in fat extracted from French fries before and after one, five, and ten days of frying is presented in Figure 2.

The total content of sterols in the fat extracted from French fries before frying amounted 2.0 mg/g, and the levels of β-sitosterol, campesterol, and avenasterol were, respectively, 61%, 28%, and 7%. The percentage of brassicasterol and stigmasterol was 2% for both sterols. Similar total sterol contents were found in the fats extracted from French fries after frying in HOSO for one day. The composition of this fraction was different, with β-sitosterol, campesterol, stigmasterol, brassicasterol, and avenasterol at, respectively, 57%, 26%, 10%, 5%, and 3%.

The greatest increase in total sterol content was found after frying French fries in HOLLRO for five and ten days, followed by French fries after one day of frying. The percentage composition of all the fats was similar and was 10–11% brassicasterol, 26–29% campesterol, 57–58% β-sitosterol, and 1 and 3% stigmasterol and Δ5-avenasterol, respectively.

The total sterol content in the fat extracted from French fries after frying in HOSO for five and ten days was 150% higher than after one day of frying. The increase in brassicasterol content from 0.04 mg/g in the French fries prior to frying to 0.2 mg/g after five and ten days of frying may show that French fries absorb sterols from the frying medium. Sterols from the prefried battered chicken and fish moved to the fried medium, and as a result of rotational frying, were absorbed by the French fries.

Using SO for frying led to an increase in the total sterol content of French fries after five days of frying, while frying in HORO lead to an increase after the first day of frying. After day ten of frying in SO, the sterol level and composition remained unchanged. When HORO was used, the content and composition of sterols after ten days was the same as after the first day of frying.

The data show that using HOLLRO for rotational frying of French fries gave the highest sterol levels, and this did not depend on the duration of frying.

### 3.3. Prefried Battered Chicken

The total sterol content and composition in prefried battered chicken before and after frying for one, five, and ten days is presented in Figure 3. The highest total sterol levels of 6.0 mg/g were noted in the chicken prior to frying. The main sterol was β-sitosterol, followed by cholesterol, campesterol, brassicasterol, avenasterol, and stigmasterol; these amounted to 41%, 30%, 19%, 7%, 2%, and 1%, respectively.

The lowest level of sterol degradation was found in prefried chicken fried in HORO, where a total sterol level of 5.3–5.6 mg per gram of oil was determined. The percentage composition was also similar to that before frying, with β-sitosterol as the main component at 41–43%, followed by cholesterol at 25–29%, and campesterol at 22–23%. The quantities of brassicasterol, avenasterol, and stigmasterol remained the same and amounted to 5%, 3%, and 1% for all the chicken samples fried in HORO. When HOLLRO was used for frying, the fat extracted from the chicken contained 4.9 to 5.1 mg of sterols per gram. As with the previous oil, the sterol fraction consisted mainly of β-sitosterol at 41%, cholesterol at 26–28%, and campesterol at 20–22%. Brassicasterol accounted for 7–8%, avenasterol for 2–3%, and stigmasterol for 1%.

The composition and content of sterols in the fat extracted from chicken fried in soybean oils (SO and HOSO) differed significantly from the case where rapeseed oils were used. After the first day of frying, the fat extracted from the chicken contained 3.6 mg of sterols in 1 g; after five days this decreased to 2.5 mg/g, and after the tenth day it increased to 3.6 mg/g. The percentage of β-sitosterol ranged from 35% after the first day, to 42% after the fifth day, and to 37% after ten days. The cholesterol share was 39% of the sterol fraction after the first day of frying, but after five and ten days it decreased to 29%. Campesterol was the third most common sterol; its proportion increased during frying, from 16% to 23%.

Our data show that when chicken was fried in both rapeseed oils, cholesterol was observed to degrade more than plant sterols (Figure 4). The degradation of cholesterol ranged from 12% to 28%, and that of phytosterols from 0.6% to 15%.

The greatest decrease in sterols after the first day of frying was seen in the chicken fried in HOSO, with sterol levels decreasing from 6.0 mg/g to 2.4 mg/g. After the fifth and tenth days, the sterol levels increased to 4.6 and 4.3 mg/g, respectively—much higher than for SO but close to the values for HOLLRO.

Crucially, the degradation of cholesterol and total plant sterols (except for SOCh1) was similar when the chicken was fried in in either soybean oil, but greater than when fried in either rapeseed oil (Figure 4).

### 3.4. Prefried Fish Sticks

Figure 5 shows the total sterol content and composition in prefried fish sticks before frying and after frying for one, five, and ten days. The greatest amount of total sterols, at 6.5 mg/g, was noted for fish sticks prior to frying. The main sterol was β-sitosterol, followed by cholesterol, campesterol, brassicasterol, avenasterol, and stigmasterol. These amounted to 38%, 29%, 24%, 6%, 3%, and 1%, respectively.

After one day of frying fish sticks in HORO, the sterol content decreased to 3.6 mg/g, after five days it reached 3.4 mg/g, and after ten days 2.5 mg/g. The main sterol was β-sitosterol, which made up 41–45% of the sterol fraction. The percentages of campesterol and cholesterol were similar at 24–26% and 20–27%, respectively. The percentage of brassicasterol did not change, being 6% regardless of frying time. Avenasterol accounted for 2–3%. Stigmasterol was not determined as it fell below the detection threshold. The total decrease in sterols was 48–61%. The amount of cholesterol that had degraded in the fat extracted from the fish sticks after the first day of frying was 54%; after the fifth day it reached 51% and after the tenth 73%. The total phytosterols decreased by 41%, 46% and 56% over the same periods (Figure 6). This high sterol degradation level was not seen in any of the tested products rotationally fried in this oil.

When fish sticks were fried in HOLLRO, the extracted fat contained 2.3–2.6 mg of sterols per gram. The percentage of β-sitosterol was 41–44%, and this was followed by cholesterol (23–27%), campesterol (20–23%), brassicasterol (8–9%), and avenasterol (2% regardless of frying duration). Stigmasterol was not detected, just as for fish fried in HORO. 

The greatest level of sterol degradation was detected in fat extracted from fish sticks fried for one day in SO; the sterol content of the extracted fats was 1.6 mg/g. After the fifth and the tenth day of frying, this increased to 1.9 and 3.5 mg/g, respectively. Cholesterol content increased from 0.5 mg/g after the first day of frying to 1.1 mg/g after the tenth day of frying. 

When HOSO was used to fry fish sticks, sterol degradation was lower than during frying in SO, but still higher than for HORO or HOLLRO. After frying for one day, the sterol content was 2.1 mg/g; this decreased to 1.7 mg/g after five days and to 1.9 mg/g after ten days (Figure 5). The percentage of total phytosterol degradation was 72–73% regardless of frying duration, but the cholesterol decreased 57%, 76%, and 69% after one, five, and ten days of frying, respectively (Figure 6).

## 4. Discussion

This study has, for the first time, shown the effects of rotational frying of three different food products (French fries, battered chicken, and fish sticks) on the degradation of plant sterols and cholesterol and on their migration between the oil and the fried food. High temperatures are typically used during frying, resulting in chemical reactions such as oxidation, hydrolysis, polymerization/polycondensation, isomerization, and cyclization [18]. Many authors have described the changes that occur in vegetable fat and oil quality during heating or frying process of a single product, such as French fries [19,20], fish [21,22], or chicken [23,24], reporting changes in a range of parameters and constituents, including fat content, color, fatty acids, polar compounds, anisidine values, and acid values. Much less information can, however, be found on changes in sterols during frying. Thermo-oxidative degradation of sterols has been measured in different vegetable fats and oils heated to 180 °C to simulate the frying process [25,26]. The data from these studies demonstrate the degradation of individual and total sterols during heating in this model system. The addition of food products to heated fats and oils significantly changes the overall system, resulting in a much wider range of reactions. Most of the available literature describes the changes that take place in fat when frying a single food product, and only a few works give any information on the interaction of sterols from the frying medium with sterols from the food product.

When French fries were fried in high-oleic sunflower oil, the percentage of linoleic acid had a greater effect on oil stability than the tocopherol content did [27]. The effect of temperature on the PUFAs in French fries fried in canola oil was described by Aladedunye and Przybylski [19]. Information specifically on the changes in fish and fish products upon frying at different frying cycles, times, temperatures, oils, and techniques was reviewed by NurSyahirah and Rozzamri [28], but changes in sterols are not described there. To decrease the detrimental effect of the frying process on the quality of the oil and chicken nuggets, vacuum frying and using the lowest possible frying temperature has been suggested [24]. These authors did not examine the oxidative degradation of cholesterol in chicken nuggets and plant sterols in oils.

There are very limited literature data on sterol degradation and migration between the frying media such as vegetable oils and prefried food products. Ozogul et al. [29] described a remarkable increase in sitosterol (more than 3–4 fold compared to raw fish) for some fish species cooked in the oven. Interestingly, a spent layer of chicken nuggets had a remarkably low content of cholesterol in comparison to other chicken meat types [30], which was in agreement with the result of Gonçalves Albuquerque et al. [31] that the cholesterol content of baked chicken was twice as high as that of fried chicken.

The effect of the frying medium and the type of food products on the quality of lipids in fried food was analyzed using ^1^H nuclear magnetic resonance by Martinez-Yusta and Guillén [32]. They demonstrated that the nutritional and health properties of fried food depend most strongly on the thermo-degradation of the frying medium, followed by the original lipid content of the food and the capacity of the food to absorb the frying medium. The concentration of β-sitosterol and Δ5-campesterol in doughnuts was greater than in the corresponding frying media derived from the three oils [32]. The transfer of bioactive compounds from virgin olive oil to deep-fried French fries showed that sterols presented the highest transfer rate and that bioactive compounds were particularly transferred during the first two days of frying [12]. French fries were subjected to successive frying for 24 h in a mixture of soybean oil, linseed oil, and safflower oil and the data showed a higher level of linolenic acid but a greater degradation of sterols than in French fries fried in soybean oil [33].

Our results show that cholesterol from fried chicken and fish sticks was not transferred to fried oils or French fries. The level of sterol degradation during frying depended on the kind of oil. HOSO showed higher sterol degradation during frying than HORO, while for French fries and chicken, HOLLRO was the best of all the oils we tested.

The literature contains data on the effects of rotational frying for up to twelve days on fatty acid composition, polar components, tocopherols, anisidine values, color, and sensory assessments [14]. The high-oleic low-linolenic canola oil exhibited the greatest frying stability in terms of polar component, oligomer, and nonvolatile carbonyl component formation. Additionally, food products fried in the high-oleic low-linolenic canola oil received the best scores in a sensory acceptance assessment [14]. These data are in agreement with our results: we have shown that sterols were most stable when food products were fried in HOLLRO.

The effect of the quality of the fat used for prefrying the tested products cannot be overlooked when considering our results. During frying, this fat passes into the medium and can be absorbed by other products fried afterwards in the medium. In restaurant practice, the quality of the fat used for prefrying tends to be overlooked. Directly frozen products are prefried in fat heated to 180–190 °C, which also introduces degradation products from prefrying. Our results have, for the first time, shown the effect of the type of oil used on sterol degradation in both oils and food products. Sterol degradation in products fried in HOLLRO was the lowest. However, it should be noted that the lowest sterol content was recorded in fish, despite the fact that the sterol content of this product prior to frying was the highest. It is likely that the fish was prefried in oil with a high sterol content, which then degraded very quickly, regardless of the medium used.

## 5. Conclusions

All sterols, including cholesterol and plant sterols, are thermolabile compounds that degrade during frying. The degradation products formed are toxic to humans, and their levels should be kept as low as possible. Their degree of degradation is affected by the quality of the frying medium used, the fat used for prefrying, and the type of product being fried. The rotational frying process did not lead to migration of cholesterol from animal products to the frying medium and the French fries. Supplementation of the frying medium by the addition of fresh oil has a positive effect on the level of sterols in the finished product.

HOLLRO can be recommended as the best solution for frying when the low sterol degradation and migration are considered. This also leads to an increase in the nutritional value of fried food products.

## Figures and Tables

**Figure 1 biomolecules-14-00269-f001:**
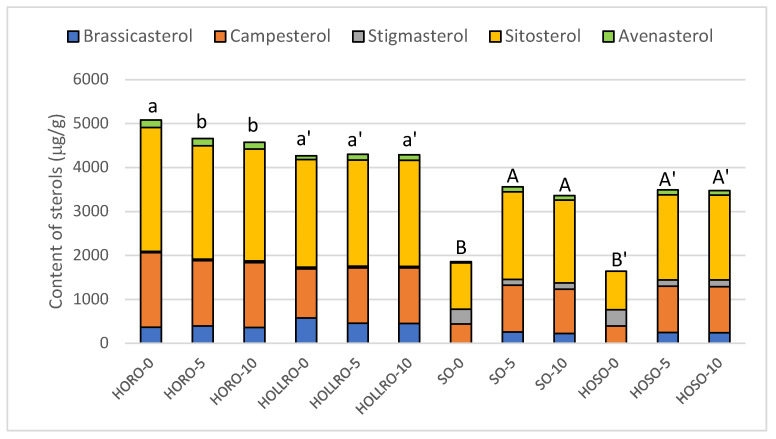
The content and composition of sterols in the oils used for rotational frying of French fries, chicken, and fish before and after five and ten days of frying. HORO: high-oleic rapeseed oil; HOLLRO: high-oleic low-linolenic rapeseed oil; SO: soybean oil; and HOSO: high-oleic soybean oil. The letters above the bars show the differences (*p* < 0.05) for the separate oils.

**Figure 2 biomolecules-14-00269-f002:**
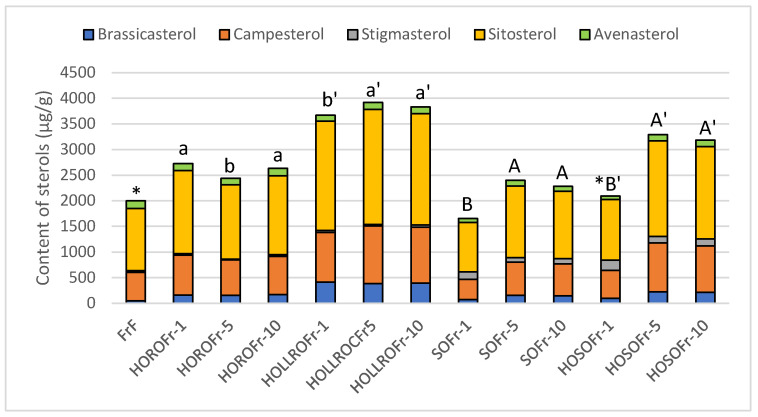
Content and composition of sterols in fats extracted from French fries before (FrF) and after rotational frying for one, five, and ten days. HORO: high-oleic rapeseed oil; HOLLRO: high-oleic low-linolenic rapeseed oil; SO: soybean oil; and HOSO: high-oleic soybean oil. The letters above the bars show differences for French fries fried in different oils (*p* < 0.05). * shows the similarity of fats from fresh French fries to other samples.

**Figure 3 biomolecules-14-00269-f003:**
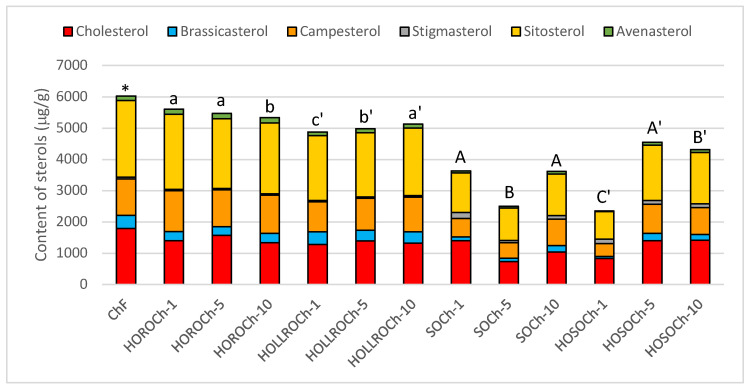
Sterol content and composition of the fats extracted from battered chicken before (ChF) and after rotational frying for one, five, and ten days. HORO: high-oleic rapeseed oil; HOLLRO: high-oleic low-linolenic rapeseed oil; SO: soybean oil; HOSO: high-oleic soybean oil. Letters above bars show differences for battered chicken fried in different oils (*p* < 0.05). * marks where fats from fresh battered chicken are similar to those from other samples.

**Figure 4 biomolecules-14-00269-f004:**
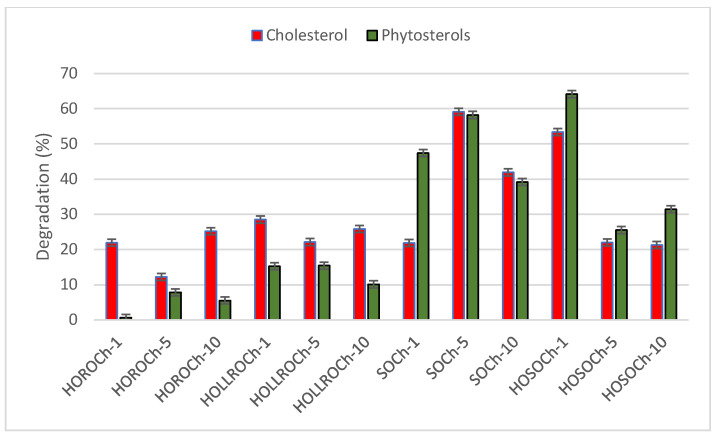
Percentage degradation of cholesterol and total phytosterols during frying of prefried chicken in oil. HORO: high-oleic rapeseed oil; HOLLRO: high-oleic low-linolenic rapeseed oil; SO: soybean oil; and HOSO: high-oleic soybean oil.

**Figure 5 biomolecules-14-00269-f005:**
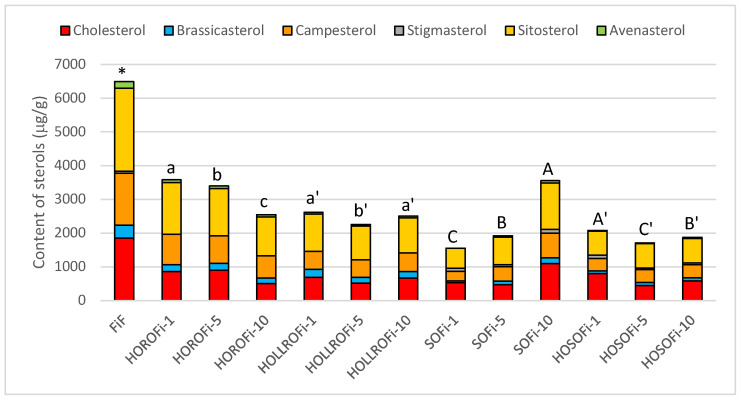
Sterol content and composition of fat extracted from fish sticks before (FiF) and after rotational frying for one, five, and ten days. HORO: high-oleic rapeseed oil; HOLLRO: high-oleic low-linolenic rapeseed oil; SO: soybean oil; HOSO: high-oleic soybean oil. Letters above bars show differences for fish sticks fried in different oils (*p* < 0.05). * marks where fats from fresh battered chicken are similar to those from other samples.

**Figure 6 biomolecules-14-00269-f006:**
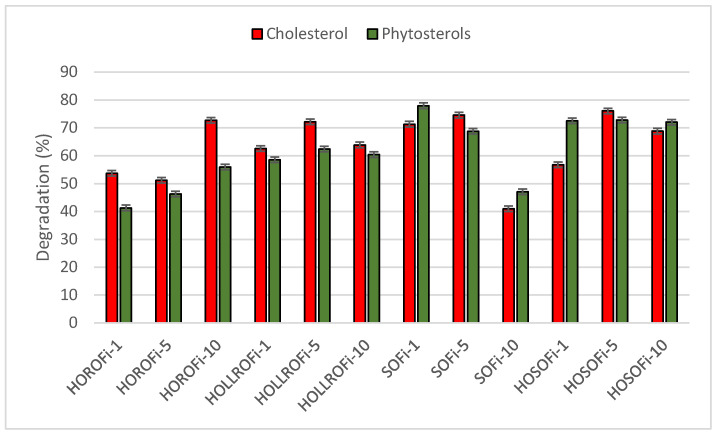
Percentage degradation of cholesterol and total phytosterols during frying of prefried fish sticks (Fi) in oil. HORO: high-oleic rapeseed oil; HOLLRO: high-oleic low-linolenic rapeseed oil; SO: soybean oil; and HOSO: high-oleic soybean oil.

## Data Availability

The research data will be available upon request to the corresponding author.

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
