# Peer review of "Sterol Migration during Rotational Frying of Food Products in Modified Rapeseed and Soybean Oils"

_biomolecules, 2024, doi:10.3390/biom14030269_

Round 1
Reviewer 1 Report
Comments and Suggestions for Authors
The manuscript entitled “Sterol migration during rotational frying food of products in modified rapeseed and soybean oils” is an interesting contribution to study the effect of rotational frying food on the sterol migration. The introduction is well written, objective is clear, conclusions are supported by data, only minor details are needed before accepting the manuscript.
Comments
Abstract
Please define acronymus “ HOLLRO. HOSO and HORO”
Introduction
Ok
Lines 92- 94 Please add more details if available about prefried conditions of food products
“Frozen French fries that had been prefried in high-oleic low-linolenic rapeseed oil and battered chicken and fish sticks prefried in refined rapeseed oil were used in the frying experiments.”
Why do not used method to evaluate degree oxidation of the used oil? (i.e p-anisidine, free fatty acids, conjugated dienes, etc)
Water content in the oils was not evaluated? Why?
Author Response
Reviewer1
Dear Reviewer,
thank you very much for all your comments and valuable suggestions. I have included the responses below. I have marked in red the longer texts added in the manuscript.
Abstract
Please define acronymus “ HOLLRO. HOSO and HORO”
- Acronymus HOLLRO – high oleic low linolenic rapeseed oil; HOSO – high oleic soybean oil; HORO – high oleic rapeseed oil used in abstract have been corrected.
Introduction
Ok
Lines 92- 94 Please add more details if available about prefried conditions of food products
“Frozen French fries that had been prefried in high-oleic low-linolenic rapeseed oil and battered chicken and fish sticks prefried in refined rapeseed oil were used in the frying experiments.”
- The same batch of frozen French fries, battered chicken and fish sticks used in experiment were purchased in local markets (Lethbridge, Canada).
- Unfortunately, we do not have exact data on the frying process of the products used in our experiment. These products were purchased from local markets because we wanted to observe the changes in sterols during the frying of such products. They are often delivered to the consumer or restaurant, where they are fried in the same oil, in rotation - French fries, meat and fish.
Why do not used method to evaluate degree oxidation of the used oil? (i.e p-anisidine, free fatty acids, conjugated dienes, etc)
- This publication deals only with sterol transformations during frying. Determination of anisidine acid number and other indicators such as tocopherols, polar compounds, fatty acids were previously published. In contrast, there are no data in the literature on sterols.
Water content in the oils was not evaluated? Why?
- Unfortunately, we did not determine the water content of the fried oils as we focused on sterol degradation. But this is an important consideration and we will definitely include these determinations in future studies. Thank you for this suggestion.

Reviewer 2 Report
Comments and Suggestions for Authors
In this work, the authors studied how rotational frying of three different food products (fries, chicked and fish) affects sterol degradation, as well as transfer between them and the frying oil, over several rounds of frying spanning around 10 days. Four types of frying oils were investigated: high-oleic rapeseed oil (HORO), high-oleic low-linolenic rapeseed oil (HOLLRO), soybean oil (SO) and high-oleic soybean oil (HOSO). Sterol degree of degradation was shown to be effected by the quality of the frying medium used, the fat used for prefrying, and the type of product being fried. The rotational frying process did not lead to migration of cholesterol from animal products to the frying medium and the French fries. Authors also showed that supplementation of the frying medium by the addition of fresh oil had a positive effect on the level of sterols in the end product.
The idea of studying rotation frying effects in re-used oil and its consequences on loss of valuable phytosterols and/or transfer between animal and vegetable sources seems original and worthy of reporting in this journal.
I have some recommendations for filling some gaps and for improving the quality of the current version of the manuscript:
Title: incorrect word order, should be "rotational frying of food products"
Line 13: impact ... on (not "regarding")
Line 13: its migration (not "their", since it is about the sterol)
Line 20-21: what are the meanings of these acronyms ? HOLLRO, HOSO, HORO. They must be defined where they first occur in the manuscript i.e. here.
Line 22: rephrase to "Despite having initially the highest sterol content"
Line 46: plays (singular)
Line 72: What is CLA? Please define your acronyms
Line 78: remarkably (adverb) instead of remarkable (adjective)
Line 82: frying OF products
Line 87: please be more specific here. Study what exact aspects of cyclic frying?
Section 2.1 of Materials and Methods: These are too vague descriptions for other researchers to be able to reproduce your work and gain the same results. Please be more specific: type of potato/species, what kind of chicken and what kind of fish, producers, where did you get every kind of oil?
Line 115: Indicate freezer type and manufacturer
Line 127: what kind of retention "data" are you referring to ? Retention time or retention index or both?
Section 2.5: 5alpha-Cholestane is mentioned to have been used as internal standard. If quantification of other sterols was done by peak area comparison with that of this standard, then relative response factors of all sterols with respect to this internal standard should have been determined and must be reported in the paper. Or external calibration curve equations must be given with their linear ranges if external calibration was used.
Line 141-142: This discussion compares stigmasterol content between different types of oil; however, the letter symbols used in Figure 1 only mark comparisons with respect to the time-course variation of composition for each type of oil in part. Full statistical comparisons between all possible groups must be made and marked correspondingly on the graph in Fig. 1 Similar comment for the other figures that follow, where this is the case.
Line 334: replace the symbol "&" with "and"
Line 337: rephrase to "lipid content of the food and the capacity of food to absorb"
Conclusion: it is merely a summary right now; add also a phrase or two about the impact of your research findings in practice and some guidelines or recommendations for frying procedures that can be formulated based on your results herein
Comments on the Quality of English Language
Minor edits needed
Author Response
Reviewer 2
Dear Reviewer,
thank you very much for all your comments and valuable suggestions. I have included the responses below. I have marked in red the longer texts added in the manuscript.
In this work, the authors studied how rotational frying of three different food products (fries, chicked and fish) affects sterol degradation, as well as transfer between them and the frying oil, over several rounds of frying spanning around 10 days. Four types of frying oils were investigated: high-oleic rapeseed oil (HORO), high-oleic low-linolenic rapeseed oil (HOLLRO), soybean oil (SO) and high-oleic soybean oil (HOSO). Sterol degree of degradation was shown to be effected by the quality of the frying medium used, the fat used for prefrying, and the type of product being fried. The rotational frying process did not lead to migration of cholesterol from animal products to the frying medium and the French fries. Authors also showed that supplementation of the frying medium by the addition of fresh oil had a positive effect on the level of sterols in the end product.
The idea of studying rotation frying effects in re-used oil and its consequences on loss of valuable phytosterols and/or transfer between animal and vegetable sources seems original and worthy of reporting in this journal.
I have some recommendations for filling some gaps and for improving the quality of the current version of the manuscript:
Title: incorrect word order, should be "rotational frying of food products"
- The title has been corrected.
Line 13: impact ... on (not "regarding")
- It has been corrected.
Line 13: its migration (not "their", since it is about the sterol)
- The paper is about the migration of different sterols so the 's' was added to the word 'sterol' and the 'their' was left in.
Line 20-21: what are the meanings of these acronyms ? HOLLRO, HOSO, HORO. They must be defined where they first occur in the manuscript i.e. here.
- It has been corrected.
Line 22: rephrase to "Despite having initially the highest sterol content"
- This sentence has been corrected.
Line 46: plays (singular)
- It has been corrected.
Line 72: What is CLA? Please define your acronyms
- The abbreviation of CLA was explained.
Line 78: remarkably (adverb) instead of remarkable (adjective)
- It has been corrected.
Line 82: frying OF products
- It has been corrected.
Line 87: please be more specific here. Study what exact aspects of cyclic frying?
- The main goal and cyclic frying have been described more detailed.
- The main goal of the present work was to study the sterol migration and degradation during rotational frying of French fries, battered chicken, and fish sticks in high-oleic rapeseed oil (HORO), high-oleic low-linolenic rapeseed oil (HOLLRO), soybean oil (SO) and high-oleic soybean oil (HOSO). Prefried French fries, battered chicken and fish stocks were fried succession forming one rotational cycle and nine cycles were run daily in each oil.
Section 2.1 of Materials and Methods: These are too vague descriptions for other researchers to be able to reproduce your work and gain the same results. Please be more specific: type of potato/species, what kind of chicken and what kind of fish, producers, where did you get every kind of oil?
- Producers of food products and oils have been added.
- The same batch of frozen French fries, battered chicken and fish sticks used in experiment were purchased in local markets (Lethbridge, Canada).
- All oils were commercially refined, bleached and deodorized and obtained from Richardson Oil Processing (Lethbridge, Canada).
Line 115: Indicate freezer type and manufacturer
· It was freezer Whirlpool - WSZ57L18DM.
Line 127: what kind of retention "data" are you referring to ? Retention time or retention index or both?
- We used both methods for identification of sterols in our samples.
Section 2.5: 5alpha-Cholestane is mentioned to have been used as internal standard. If quantification of other sterols was done by peak area comparison with that of this standard, then relative response factors of all sterols with respect to this internal standard should have been determined and must be reported in the paper. Or external calibration curve equations must be given with their linear ranges if external calibration was used.
- The relative response factors for all sterols ranged from 0.985 to 1.010, so an RRF of 1.0 was used in all peak area conversions.
Line 141-142: This discussion compares stigmasterol content between different types of oil; however, the letter symbols used in Figure 1 only mark comparisons with respect to the time-course variation of composition for each type of oil in part. Full statistical comparisons between all possible groups must be made and marked correspondingly on the graph in Fig. 1 Similar comment for the other figures that follow, where this is the case.
- Canola oils have a very low stigmasterol content, while this level is much higher in soybean oils. This was pointed out when describing the differences in sterol composition in the oils studied, which could then affect their stability. Stigmasterol has two double bonds in the molecule and its degradation can occur faster than other sterols. We believe that additional statistical analysis would have forced a change in the type of graph, which at the same time would have affected its readability. The stability of stigmasterol in a model system was the subject of my earlier publication.
- RudziÅ„ska, M., Korczak, J., Gramza, A., WÄ…sowicz, E., Dutta, P.C. Inhibition of stigmasterol oxidation by antioxidants in purified sunflower oil. Journal of AOAC International, 2004, 87(2), pp. 499–504
Line 334: replace the symbol "&" with "and"
- It has been corrected.
Line 337: rephrase to "lipid content of the food and the capacity of food to absorb"
- It has been corrected.
Conclusion: it is merely a summary right now; add also a phrase or two about the impact of your research findings in practice and some guidelines or recommendations for frying procedures that can be formulated based on your results herein
- HOLLRO can be recommended as the best solution for frying when decrease of sterols degradation and migration is considered. This also give the increase of nutritional value of fried food products.